# Diffusion Measures of Subcortical Structures Using High-Field MRI

**DOI:** 10.3390/brainsci13030391

**Published:** 2023-02-24

**Authors:** Hyeon-Man Baek

**Affiliations:** 1College of Medicine, Gachon University, Incheon 21565, Republic of Korea; hmbaek98@gachon.ac.kr; Tel.: +82-010-9878-4279; 2Department of Health Sciences and Technology, GAIHST, Gachon University, Incheon 21999, Republic of Korea

**Keywords:** Parkinson’s disease, basal ganglia, diffusion MRI, 3T, 7T

## Abstract

The pathology of Parkinson’s disease (PD) involves the death of dopaminergic neurons in the substantia nigra (SN), which slowly influences downstream basal ganglia pathways as dopamine transport diminishes. Diffusion magnetic resonance imaging (MRI) has been used to diagnose PD by assessing white matter connectivity in some brain areas. For this study, we applied Lead-DBS to human connectome project data to automatically segment 11 subcortical structures of 49 human connectome project subjects, reducing the reliance on manual segmentation for more consistency. The Lead-connectome pipeline, which utilizes DSI Studio to generate structural connectomes from each 3T and 7T diffusion image, was applied to 3T and 7T data to investigate possible differences in diffusion measures due to different acquisition protocols. Significantly higher fractional anisotropy (FA) values were found in the 3T left SN; significantly higher MD values were found in the 3T left SN and the right amygdala, SN, and subthalamic nucleus (STN); significantly higher AD values were found in the right RN and STN; and significantly higher RD values were found in the left RN and right amygdala. We illustrate a methodology for obtaining diffusion measures of basal ganglia and basal ganglia connectivity using diffusion images, as well as show possible differences in diffusion measures that can arise due to the differences in MRI acquisitions.

## 1. Introduction

The basal ganglia have been an important site for investigation in studying diseases such as Parkinson’s disease. Structures such as the substantia nigra (SN), where death of dopaminergic neurons are reported in Parkinson’s disease, are frequently investigated for changes in shape and various characteristics that could provide explanations for disease pathology and symptoms [1]. Investigations of the basal ganglia motor circuit has led to conclusions as to how disruption of dopamine from SN affects downstream pathways that rely on dopamine for motor signal regulation [2]. As such, various treatments for Parkinson’s disease have been focused on trying to supplement alternatives to make up for dopamine reduction in the system, such as L-dopa injection and deep brain stimulation (DBS) [2,3]. Lead-DBS is an open-source toolbox designed for providing precise locations of anatomical structures and electrode placement for deep brain stimulation, to segment key structures involved in Parkinson’s disease. Lead-DBS provides multitude of automated atlas-based segmentation methods for segmenting regions and subregions of the brain not commonly done by other open-source toolboxes. However, due to the deep location of the basal ganglia in the brain, the methodology used to document the effects of Parkinson’s disease treatments such as MRI has undergone multiple steps in improvement to increase the signal-to-noise ratio for visualizing the basal ganglia. Such improvements have also positively affected diffusion MRI acquisitions, which are used to simulate white matter connectivity for studying how disease and treatments affect the white matter tracts that connect SN to other structures of the basal ganglia.

Estimating the connectivity of white matter fiber pathways involves methodologies such as diffusion tractography to sample streamlines from fiber orientations calculated within each voxel of a diffusion image. To achieve optimal results with diffusion tractography, the source diffusion data must have high enough spatial resolution to reduce the ambiguity in where fiber crossings occur. For example, voxels with large voxel sizes could contain crossing fiber orientations that would be difficult to differentiate, regardless of the location of the voxel in which the crossing occurs (crossing fibers in a “T” shape would show the same signal as a “+” shaped crossing fiber) [4]. However, improving spatial resolution comes at a cost of lost signal-to-noise ratio (SNR), making it difficult to achieve optimal spatial resolution for fitting fiber orientations. Use of higher field MRI (7T) has shown that it is possible to obtain higher spatial resolution than what is offered at 3T MRI with acceptable SNR, demonstrating the potential for improvements in diffusion data [5].

The Human Connectome Project (HCP), a consortium aimed at characterizing brain connectivity of 1200 healthy adults, incorporated both 3T and 7T acquisitions of 200 subjects. While 7T diffusion acquisitions were acquired with higher spatial resolution than 3T diffusion acquisitions (1.05 mm for 7T, 1.25 mm for 3T), various factors regarding inhomogeneities and protocols may influence the consistency of fiber orientation estimation in 7T diffusion weighted images, more so than in 3T diffusion weighted images. B0 inhomogeneities resulting in distortion and non-uniformity of signal intensity scale with magnetic field strength, which may result in inconsistent tractography in areas with dropped signals [6]. Additionally, a weaker gradient set was used for 7T acquisitions (70 mT/m for 3T, 100 mT/m for 7T) along with lower angular contrast and angular resolution, possibly offsetting the SNR gains from increased field strength [7,8]. HCP data were preprocessed through the HCP pipeline, which corrects for intensity normalization and distortion; however, there still may be minor inconsistencies that could influence diffusion tractography.

For this study, we compare 3T and 7T diffusion measures generated between 11 subcortical structures: the amygdala, caudate, globus pallidus externus (GPe), globus pallidus internus (GPi), hippocampus, nucleus accumbens (NA), putamen, red nucleus (RN), subthalamic nucleus (STN), substantia nigra (SN), and thalamus. In order to do so, we use Lead-DBS [9], an open-source toolbox with a multitude of automated atlas-based segmentation methods and the DSI Studio tractography software tool for diffusion MRI analysis [10,11,12]. We aimed to describe a methodology for obtaining diffusion measures of subcortical structures such as fractional anisotropy (FA), mean diffusivity (MD), axial diffusivity (AD), radial diffusivity (RD), and quantitative anisotropy (QA). We also investigated if there were possible significant differences in diffusion measures of connectivity tracts representing pathways due to differences in the acquisition protocol, such as field strength differences.

## 2. Materials and Methods

### 2.1. Subjects

Forty-nine female subjects from the WU-Minn HCP Young Adult 1200 Subjects dataset were used for this study [13]. All subjects were young adults (age range: 22–35 years) and healthy, with no documented neuropsychiatric disorders, neurologic disorders, or illnesses such as diabetes and high blood pressure. The data used in this study were appropriately obtained through methods approved by an institutional review board and were HIPPA-compliant. Informed consent was obtained from each patient prior to the study.

### 2.2. MRI Acquisition

MR scans were performed in a 3T Siemens Skyra scanner and a 7T Siemens MAGNETOM, equipped with a 100 mT/m and a 70 mT/m gradient set using a 32-channel head coil, respectively [14].

Three-dimensional MPRAGE sequences for T1-weighted images were used at 3T. The scan parameters were: echo time (TE) = 2.14 ms, repetition time (TR) = 2400 ms, inversion time (TI) = 1000 ms, flip angle (FA) = 8o, field of view (FOV) = 180 × 224 × 224 mm^3^, voxel size = 0.7 mm isotropic, bandwidth (BW) 210 Hz/Px, acquisition time = 7 min 40 s.

Three-dimensional T2-SPACE sequences for T2-weighted images were used at 3T. The scan parameters were: echo time (TE) = 565 ms, repetition time (TR) = 3200 ms, field of view (FOV) = 180 × 224 × 244 mm^3^, voxel size = 0.7 mm isotropic, bandwidth (BW) = 744 Hz/Px, acquisition time = 8 min 24 s.

Multiband spin-echo EPI sequences for diffusion weighted images were used at 3T. The scan parameters were: echo time (TE) = 89.5 ms, repetition time (TR) = 5520 ms, flip angle (FA) = 78°, refocusing flip angle (rFA) = 160°, field of view (FOV) = 210 × 180 (RO × PE), matrix = 168 × 144 (RO × PE), slice thickness = 1.25 mm, 11 slices, 1.25 mm isotropic voxels, multiband factor = 3, echo spacing = 0.78 ms, bandwidth (BW) = 1488 Hz/Px, b-values = 1000, 2000, 3000 s/mm^2^, 90 diffusion weighting directions plus 6 b = 0 acquisitions interspersed throughout each run, acquisition time = 6 runs, one for each LR/RL direction and shell, each approximately 9 min 50 s. Total scanning time was ~55 min.

Multiband spin-echo EPI sequences for diffusion weighted images were used at 7T. The scan parameters were: echo time (TE) = 71.2 ms, repetition time (TR) = 7000 ms, flip angle (FA) = 90°, refocusing flip angle (rFA) = 180°, field of view (FOV) = 210 × 210 mm (RO × PE), matrix = 200 × 200 (RO × PE), slice thickness = 1.05 mm, 132 slices, 1.05 mm isotropic voxels, multiband factor = 2, echo spacing = 0.82 ms, bandwidth (BW) = 1388 Hz/Px, b-values = 1000, 2000 s/mm^2^, 65 diffusion weighting directions plus 6 b = 0 acquisitions interspersed throughout each run, acquisition time = 4 runs, one for each LR/RL direction and shell, each approximately 9 min 50 s. Total scanning time was ~40 min.

All images were minimally preprocessed beforehand through the HCP minimal preprocessing pipeline [14]. Through the pipeline, the T1w and T2w images were corrected for MR gradient-nonlinearity-induced distortions and readout distortions, and then bias field corrected. The diffusion weighted images were normalized for b0 image intensity, then corrected for EPI distortions, eddy-current-induced distortions, subject motion, and gradient nonlinearity. The diffusion weighted images were also resampled into 1.25 mm native structural space. More details on acquisition parameters and the preprocessing pipeline can be found in the WU-Minn HCP 1200 Subjects Data Release: Reference Manual, available at https://www.humanconnectome.org/, accessed on 21 July 2017.

### 2.3. Image Processing

To minimize differences in shells and spatial resolution between 3T and 7T diffusion acquisitions, shells with b = 3000 were removed from 3T diffusion images using dwiextract and 7T diffusion images were resampled from 1.05 to 1.25 mm using mrgrid [15].

Segmentation of the amygdala, caudate, globus pallidus externus (GPe), globus pallidus internus (GPi), hippocampus, nucleus accumbens (NA), putamen, red nucleus (RN), subthalamic nucleus (STN), substantia nigra (SN), and thalamus was performed through the default Lead-DBS pathway, excluding electrode localization and reconstruction [16]. First, a bias-field correction step using the N4 algorithm was applied to T1w and T2w images [17]. Following the bias correction, the T2w images, as well as the FA images from diffusion images, were co-registered to T1w images using SPM 12 (https://www.fil.ion.ucl.ac.uk/spm/software/spm12, accessed on 1 October 2014). Each co-registration was visually inspected using edge-detection-based wire frames overlaid on each co-registered image generated by Lead-DBS. The co-registered images were then normalized to MNI 2009b template space using advanced normalization tools (ANTs) symmetric image normalization (SyN) [18]. Then, the MNI PD25 atlas, which consists of primary target regions defined on a specialized Parkinson’s disease specific template, was warped back to each subject’s diffusion space through inverse normalization [19]. The results of segmentations were visually inspected by overlaying each inverse normalized area of interest on b0 images.

### 2.4. Diffusion Processing

LEAD Connectome, a MATLAB-based structural connectomic analysis pipeline which utilizes DSI studio for generalized Q-ball imaging (GQI), was used to generate structural connectome for each 3T and 7T acquisition [20]. The default processing settings were used, with diffusion images reconstructed using q-sampling imaging and fiber tracking with a deterministic fiber tracking algorithm [20]. A step size of 0.5 mm, angular threshold of 75°, minimum length of 10 mm, and maximum length of 300 mm were used for generating 200,000 tracts. The generated fiber tracts were then normalized to MNI 2009b template space using SPM12 coregistration [21]. Diffusion measures (FA, MD, AD, RD) of subcortical structures as well as connectivity measures (FA, MD, QA) of tractography between each ROI were obtained using DSI studio.

### 2.5. Statistical Analysis

All statistical analyses were performed using IBM SPSS Statistics for Windows (version 22.0; IBM Corp., Armonk, NY, USA). Diffusion measures of subcortical structures were compared between 3T and 7T using Student’s *t*-test. The Benjamini–Hochberg false discovery rate (FDR) procedure was applied to diffusion measure comparisons with significance set to *p* < 0.05 to correct for false positives when conducting multiple comparisons. The same methodology using Student’s *t*-test and FDR was used to find significant differences (significance at *p* < 0.05) in diffusion measures of 3T and 7T diffusion fiber tracts between each segmentation pair.

## 3. Results

A basic flowchart of the diffusion pipeline is shown in Figure 1. MNI PD25 subcortical structures that are segmented using 49 HCP subject structural images with Lead-DBS are shown in Figure 2. The results of MNI PD25 atlas ANTS automatic segmentation of an example subject’s 3T and 7T diffusion images are overlaid on top of each respective b0 image in Figure 3.

Student’s *t*-test was used to compare each diffusion measure (FA, MD, AD, RD) of ROIs between 3T and 7T diffusion weighted images. The Benjamini–Hochberg procedure was used to control for false discovery rate (FDR), with significance set to *p* < 0.05. The left substantia nigra showed significant differences in FA, MD, and RD; the right amygdala showed significant differences in MD and RD; the right red nucleus showed significant differences in AD; and the right substantia nigra and the right subthalamic nucleus showed significant differences in MD and AD. The values of each significantly different value were higher in 3T than in 7T. Table 1 shows the means of 3T and 7T FA, MD, RD, and AD, as well as the *p*-values of each comparison.

Student’s *t*-test was also used to compare the diffusion measure (QA, FA, MD) of each connection pair between each subcortical ROI segmented with Lead-DBS. Figure 4 shows the connectivity matrices (e.g., 22 × 22) of significant mean differences of FA, MD, and QA (insignificant differences set to 0, positive value denotes higher 3T mean, while negative value denotes higher 7T mean). After controlling for FDR using the Benjamini–Hochberg procedure with significance at *p* < 0.05, 240 connections out of 484 showed significant differences in FA, 234 showed significant differences in MD, and 210 showed significant differences in QA. Out of the 240 significant FA differences, 20 showed higher FA values in 7T while 220 showed higher FA values in 3T. For the 234 significant MD differences, 20 showed higher MD values in 7T while 214 showed higher MD values in 3T. For the 210 QA significant differences, 10 showed higher QA values in 7T while 200 showed higher QA values in 3T.

## 4. Discussion

For this study, we used Lead-DBS to segment basal ganglia structures, then used DSI-studio to obtain the diffusion measures of each segmented ROI and connectivity between each ROI. In addition, we compared the differences in 3T and 7T diffusion measures of each basal ganglia segmentation to observe how diffusion measures can change due to differences in acquisition protocols. Through our methodology, we were able to observe important diffusion measures of ROI and connectomes for comparing the effects of diseases such as Parkinson’s disease. Our results revealed that the values of each diffusion measures were higher in 3T than in 7T.

The basis of PD has been linked to neurodegeneration of the dopaminergic neurons found in the substantia nigra. Loss of dopaminergic neurons results in dopamine loss, disrupting the inhibitory and excitatory inputs of dopamine in basal ganglia pathways [22]. As a result, some neurons, particularly ones that connect GPe and STN, show abnormal synaptic activation due to the lack of inhibition input from dopamine. Previous studies have shown changes in diffusion measures of ROIs and connectivity that reflect the abnormalities found in PD patients. A study investigating the change in FA of STN found significant reductions in FA in subjects with mild-to-moderate PD [23]. Another study comparing basal ganglia structures between PD patients and healthy controls showed that PD patients had significant reductions in FA of SN and MD, and RD of SN and globus pallidus [24]. A study observing motor tracts such as thalamus motor tracts and the right supplementary area–putamen tract found significant increases in FA [25].

Reports of significant decreases in FA of diffusion measures have been consistent throughout various studies, particularly in the nigrostriatal tract that connects the SN to the putamen via dopaminergic neurons [26,27,28]. RD, which measures the diffusivity perpendicular to the tract, has been historically used for detecting demyelination and inflammation. An increase in RD has been correlated with demyelinated axons, as well as increased motor and memory dysfunctions [29]. The initial decreases in RD in PD were from compensations from dopaminergic deficiency, which slowly phases out due to worsened symptoms and expended neural resources as the disease progresses, resulting in increased RD of white matter tracts [30].

With the importance of comparing diffusion measures of subcortical structures for PD diffusion analysis, we aimed to observe how differences in diffusion acquisition protocols, particularly the field strengths, could affect diffusion measures of subcortical structures as well as the structural connectomes between each structure. The 3T and 7T diffusion images obtained for this study were obtained using different acquisition protocols and hardware. While certain differences regarding hardware and protocol are difficult to reconciliate, attempts to normalize the images obtained from different field acquisitions were made by matching spatial resolution (to 1.25 mm) and b values (b = 0, 1000, 2000). We used Lead-DBS because of its capability in providing accurate and consistent results in automatically segmenting previously mentioned structures [11,12]. Our results showed that visual inspection of the registered segmentations on diffusion-weighted images had minor mismatching of boundaries. While the methodology was aimed towards utilizing minimal human input and automated processes, manual adjustment of registered masks could have improved the consistency of results (Figure 3).

Several differences in diffusion measures were observed between ROIs of 3T and 7T acquisitions. As shown in Table 1, we found significant increases in FA of the 3T left SN; significant increases in MD of the 3T left SN, right amygdala, right SN, and right STN; significant increases in AD of the right RN, right SN, and right STN; and significant increases in RD in the left SN and right amygdala. While it is difficult to determine what level of diffusion measure is accurately represented, the trend of significantly lower levels of MD, AD, and RD of 7T ROIs when compared to 3T ROIs falls in line with a similar study comparing 7T and 3T acquisitions [31]. Connectivity matrices of Figure 4 represent significant mean differences of FA, MD, and QA between 3T connectomes and 7T connectomes. Similarly to the comparison between ROIs of 3T and 7T acquisitions, most of the connections between ROIs were significantly lower in FA, MD, and QA in 7T connectomes than in 3T connectomes.

There are limitations to consider regarding our results and methodology. First, the two acquisitions using 3T and 7T field strengths were subject to different protocols with varying factors such as hardware differences and parameters that could not be completely reconciled for proper comparison. Second, the diffusion measures generated from 3T and 7T ROIs and connectomes were not tested against ground truths. As such, it is unclear from the results whether significant increases in FA, MD, and QA in 3T ROIs and connectivity better represent the ground truth.

## 5. Conclusions

Our study was able to measure the diffusion measures of subcortical ROIs and connectivity. From the measurements, we were able to compare the differences in FA, MD, AD, and RD of 3T and 7T ROIs. We found that there were significant differences in the left SN, right amygdala, right SN, right RN, and right STN when comparing diffusion measures of 3T and 7T subcortical structures. Additionally, connectivity between ROIs showed more significant increases in FA, MD, and QA in 3T diffusion images when compared with 7T diffusion images. At the time of acquiring 3T and 7T data, 7T scanners were relatively new compared to 3T scanners, meaning that there was little time for the project to experiment with custom hardware or protocols to optimize 7T protocols. Additionally, maximizing the spatial resolution of diffusion imagers is crucial to minimize voxels containing multiple fiber orientations (which could result in vague false positives for streamlines), making acquisitions with higher field strengths almost a necessity to generate the most accurate connectome possible. In the future, improvements to 7T hardware and acquisition protocols should lead to a more accurate representation of a connectome.

## Figures and Tables

**Figure 1 brainsci-13-00391-f001:**
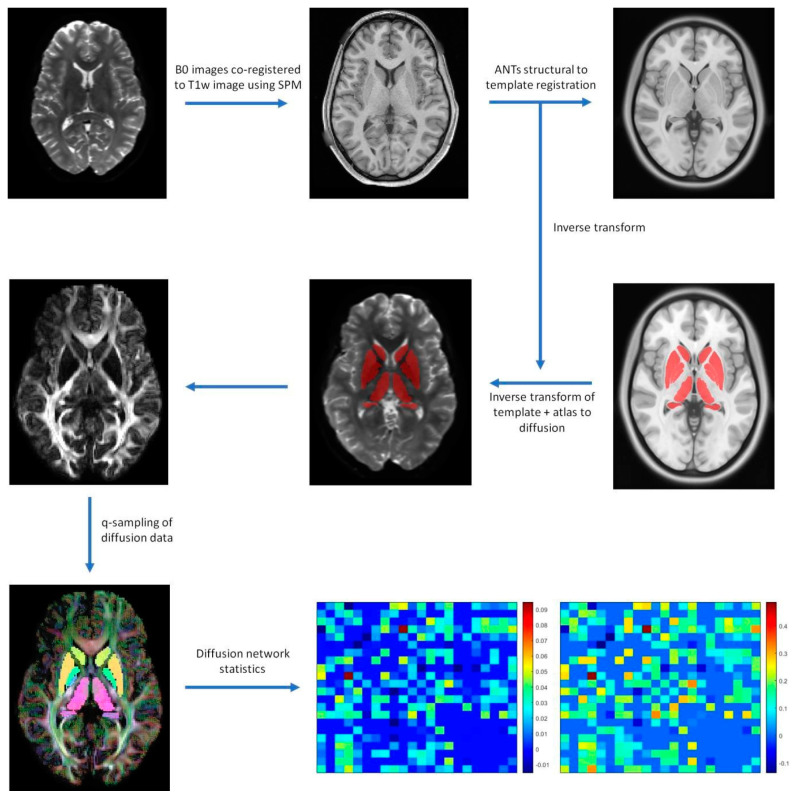
Basic processing Lead−connectome pipeline. Minimally preprocessed human connectome project images are co-registered to the structural T1w image using SPM. Structural images are registered to templates using ANTs registration, generating inverse transforms to translate MNI PD25 template to b0 space. DSI Studio is used to apply Q−sampling on diffusion images to reconstruct diffusion orientations and generate diffusion network statistics of segmented ROIs and ROI diffusion connectivity.

**Figure 2 brainsci-13-00391-f002:**
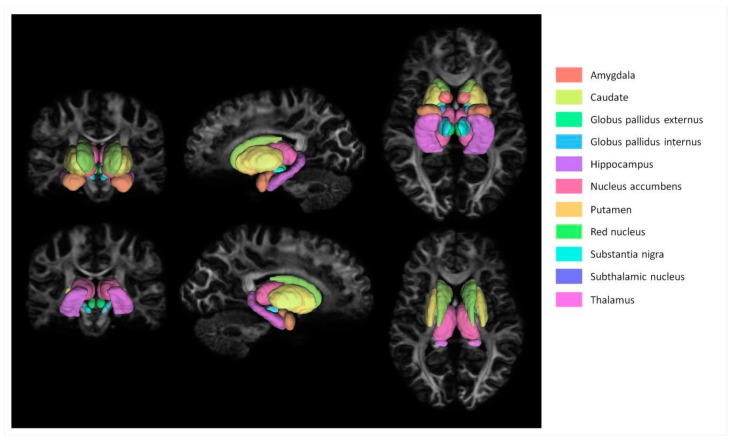
MNI PD25 atlas subcortical masks visualized over a subject’s FA image. The structures visualized in this figure were segmented using ANTs registration.

**Figure 3 brainsci-13-00391-f003:**
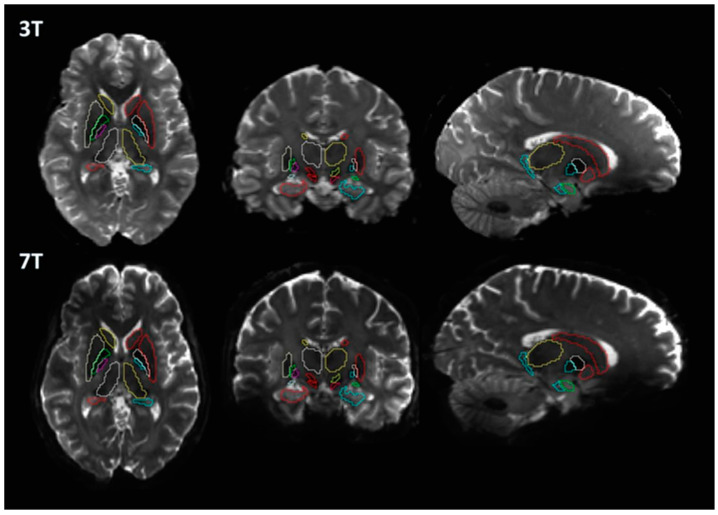
Results of 3T and 7T Lead−DBS segmentations of the same subject (HCP Subject ID: 196144) overlaid on 3T and 7T b0 images.

**Figure 4 brainsci-13-00391-f004:**
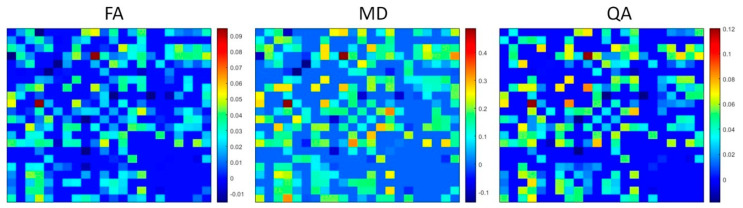
Connectivity matrices of significant, FDR−corrected mean differences of diffusion measures, FA, MD, QA, between 3T and 7T connectomes. Positive values (near red on the color bar) represent higher value of 3T mean diffusion measure while negative values (near dark blue on the color bar) represent higher value of 7T mean diffusion measure. Insignificant values were set to 0.

**Table 1 brainsci-13-00391-t001:** Significant mean differences of 11 subcortical diffusion measures (FA, MD, RD, and AD).

ROI	Mean 3T FA	Mean 7T FA	Student *T* Test *p* 3T FA > 7T FA	Mean 3T MD (×10^−3^)	Mean 7T MD (×10^−3^)	Student *T* Test *p* 3T MD > 7T MD
Left amygdala	0.189	0.184	0.645	0.748	0.669	0.098
Left caudate	0.164	0.175	0.719	0.679	0.620	0.359
Left globus pallidus externus	0.242	0.189	0.013	0.514	0.424	0.032
Left globus pallidus internus	0.250	0.196	0.597	0.525	0.433	0.890
Left hippocampus	0.168	0.157	0.351	0.788	0.712	0.632
Left nucleus accumbens	0.169	0.149	0.701	0.677	0.659	0.215
Left putamen	0.188	0.175	0.471	0.627	0.569	0.032
Left red nucleus	0.329	0.304	0.332	0.554	0.457	0.394
Left substantia nigra	**0.346**	**0.343**	**0.001**	**0.584**	**0.409**	**0.006**
Left subthalamic nucleus	0.347	0.294	0.030	0.534	0.426	0.635
Left thalamus	0.269	0.248	0.753	0.631	0.583	0.252
Right amygdala	0.169	0.175	0.509	**0.757**	**0.667**	**0.001**
Right caudate	0.177	0.161	0.017	0.677	0.608	0.748
Right globus pallidus externus	0.230	0.250	0.176	0.532	0.384	0.207
Right globus pallidus internus	0.200	0.215	0.070	0.511	0.377	0.162
Right hippocampus	0.169	0.156	0.578	0.767	0.709	0.847
Right nucleus accumbens	0.171	0.147	0.490	0.665	0.642	0.810
Right putamen	0.200	0.186	0.325	0.629	0.546	0.941
Right red nucleus	0.320	0.323	0.357	0.550	0.433	0.082
Right substantia nigra	0.480	0.400	0.743	**0.535**	**0.397**	**0.001**
Right subthalamic nucleus	0.402	0.325	0.692	**0.493**	**0.405**	**<0.001**
Right thalamus	0.272	0.254	0.595	0.636	0.573	0.472
**ROI**	**Mean** **3T AD (×10^−3^)**	**Mean** **7T AD (×10^−3^)**	**Student *T* Test *p*** **3T AD > 7T AD**	**Mean** **3T RD (×10^−3^)**	**Mean** **7T RD (×10^−3^)**	**Student *T* Test *p*** **3T RD > 7T RD**
Left amygdala	0.897	0.798	0.084	0.673	0.605	0.126
Left caudate	0.788	0.730	0.400	0.625	0.566	0.343
Left globus pallidus externus	0.636	0.500	0.043	0.452	0.386	0.033
Left globus pallidus internus	0.657	0.513	0.636	0.460	0.393	0.943
Left hippocampus	0.922	0.824	0.570	0.722	0.656	0.742
Left nucleus accumbens	0.788	0.756	0.375	0.621	0.611	0.162
Left putamen	0.751	0.675	0.040	0.565	0.516	0.090
Left red nucleus	0.751	0.605	0.018	0.455	0.383	0.985
Left substantia nigra	0.813	0.568	0.066	**0.470**	**0.329**	**<0.001**
Left subthalamic nucleus	0.717	0.560	0.067	0.442	0.359	0.749
Left thalamus	0.804	0.731	0.873	0.544	0.510	0.159
Right amygdala	0.888	0.789	0.013	**0.691**	**0.607**	**0.002**
Right caudate	0.800	0.708	0.751	0.615	0.559	0.925
Right globus pallidus externus	0.655	0.479	0.477	0.470	0.336	0.186
Right globus pallidus internus	0.612	0.450	0.800	0.460	0.340	0.065
Right hippocampus	0.898	0.821	0.925	0.701	0.654	0.788
Right nucleus accumbens	0.774	0.734	0.872	0.611	0.597	0.603
Right putamen	0.762	0.655	0.537	0.563	0.492	0.743
Right red nucleus	**0.738**	**0.581**	**0.004**	0.456	0.360	0.297
Right substantia nigra	**0.850**	**0.589**	**0.001**	0.377	0.301	0.432
Right subthalamic nucleus	**0.711**	**0.548**	**<0.001**	0.384	0.334	0.037
Right thalamus	0.814	0.721	0.132	0.547	0.498	0.861

## Data Availability

The open-source toolbox and code Lead-DBS (Horn et al., 2019) used for segmentation are available for free at www.lead-dbs.org accessed on 1 July 2015. SPM toolbox used for co-registration is available at https://www.fil.ion.ucl.ac.uk/spm/software/spm12, accessed on 1 October 2014. The toolbox and code for DSI Studio tractography software tool are available at http://dsi-studio.labsolver.org, accessed on 1 July 2015. The WU-Minn Human Connectome Project, 1200 Subjects Data Release dataset is publicly available at https://www.humanconnectome.org/, accessed on 21 July 2017.

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
