# Peer review of "Diffusion Measures of Subcortical Structures Using High-Field MRI"

_brainsci, 2023, doi:10.3390/brainsci13030391_

Round 1

Reviewer 1 Report

In this paper author has demonstrated the diffusion magnetic resonance imaging (MRI) to diagnose PD by assessing white matter connectivity in some brain areas. This paper is interesting to read but needs some explanation. My comments are given below.

What is Lead-DBS? How does Lead-DBS work? Explain briefly in the introduction section.

What are the inputs for this simulation? Draw a schematic of the procedure.

Author Response

Attached is the file below

Reviewer 2 Report

1.       I suggest a diagram to explain how you chose 11 subcortical structures of 49 human connectome project subjects with exclusion and inclusion criteria. Unfortunately, I didn´t understand very well.

2.       What does FA mean (line 16)? I think fractional anisotropy. You must write in the abstract—the same for all the abbreviations.

3.       Why did you only choose women in your study? Specify, please.

4.       Explain why you think were significant differences in Left SN, right amygdala, right SN, right RN, and right STN.

5.       Did you think that maybe the effect you found was because of the sex? Explain in the text, please.

 Best regards.

Author Response

Attached is the file below.

Reviewer 3 Report

Summary: The author presents an analysis of 3T structural and 3T and 7T diffusion MRI data from 49 healthy female subjects aged 22 - 35 years from the WU-Minn HCP Young Adult 1200 Subjects dataset. Eleven subcortical structures were automatically segmented using Lead-DBS, and then DSI Studio was used to generate the structural connectome and the deterministic tractography metrics for the segmented structures. Diffusion metrics were compared between 3T and 7T images. The authors reported a several significant, but often minor, differences between 3T and 7T data.

Major concerns:

- The introduction should include some kind of explicit statement about what this article adds to our understanding of structural connectivity between subcortical structures implicated in PD and the methodology we use to examine these connections, beyond what was published in Shim & Baek (2022) in Neuroscience. These papers are too similar to not directly address this issue throughout the entire manuscript. 

- Fig 1 is very helpful for the reader. 

- The discussion is lacking--only one paragraph (the third paragraph) attempts to place the present results within the context of what is already known in this field. Much more on this is needed. For example, only one other study is referenced in this section. 

- The beginnings of the third and fourth paragraphs in the discussion are essentially identical.

Minor concerns: 

- Line 37 - check subject-verb agreement

- Does Fig 3 use the same legend as Fig 2?

Author Response

Attached is the file below.

Round 2

Reviewer 1 Report

Authors responded my queries satisfactory. 

Reviewer 3 Report

The changes have improved the manuscript.

Please indicate in the Fig 3 caption to refer to the legend in Fig 2.